# Learning Deep Embeddings with Histogram Loss

Evgeniya Ustinova and Victor Lempitsky

Skolkovo Institute of Science and Technology (Skoltech)
Moscow, Russia

## Abstract

We suggest a loss for learning deep embeddings. The new loss does not introduce parameters that need to be tuned and results in very good embeddings across a range of datasets and problems. The loss is computed by estimating two distribution of similarities for positive (matching) and negative (non-matching) sample pairs, and then computing the probability of a positive pair to have a lower similarity score than a negative pair based on the estimated similarity distributions. We show that such operations can be performed in a simple and piecewise-differentiable manner using 1D histograms with soft assignment operations. This makes the proposed loss suitable for learning deep embeddings using stochastic optimization. In the experiments, the new loss performs favourably compared to recently proposed alternatives.

## 1   Introduction

Deep feed-forward embeddings play a crucial role across a wide range of tasks and applications in image retrieval [1, 8, 15], biometric verification [3, 5, 13, 17, 22, 25, 28], visual product search [21], finding sparse and dense image correspondences [20, 29], etc. Under this approach, complex input patterns (e.g. images) are mapped into a high-dimensional space through a chain of feed-forward transformations, while the parameters of the transformations are learned from a large amount of supervised data. The *objective* of the learning process is to achieve the proximity of semantically-related patterns (e.g. faces of the same person) and avoid the proximity of semantically-unrelated (e.g. faces of different people) in the target space. In this work, we focus on simple similarity measures such as Euclidean distance or scalar products, as they allow fast evaluation, the use of approximate search methods, and ultimately lead to faster and more scalable systems.

Despite the ubiquity of deep feed-forward embeddings, learning them still poses a challenge and is relatively poorly understood. While it is not hard to write down a loss based on tuples of training points expressing the above-mentioned objective, optimizing such a loss rarely works "out of the box" for complex data. This is evidenced by the broad variety of losses, which can be based on pairs, triplets or quadruplets of points, as well as by a large number of optimization tricks employed in recent works to reach state-of-the-art, such as pretraining for the classification task while restricting fine-tuning to top layers only [13, 25], combining the embedding loss with the classification loss [22], using complex data sampling such as mining "semi-hard" training triplets [17]. Most of the proposed losses and optimization tricks come with a certain number of tunable parameters, and the quality of the final embedding is often sensitive to them.

Here, we propose a new loss function for learning deep embeddings. In designing this function we strive to avoid highly-sensitive parameters such as margins or thresholds of any kind. While processing a batch of data points, the proposed loss is computed in two stages. Firstly, the two one-dimensional distributions of similarities in the embedding space are estimated, one corresponding to similarities between matching (*positive*) pairs, the other corresponding to similarities between non-matching (*negative*) pairs. The distributions are estimated in a simple non-parametric ways

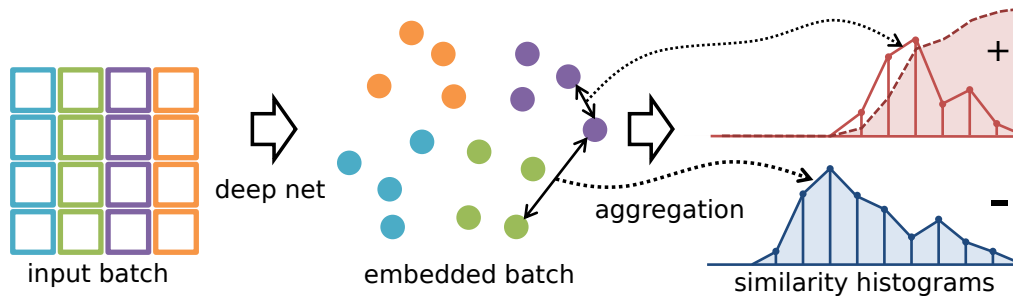

input batch     embedded batch     similarity histograms

**Figure 1:** The histogram loss computation for a batch of examples (color-coded; same color indicates matching samples). After the batch (left) is embedded into a high-dimensional space by a deep network (middle), we compute the histograms of similarities of positive (top-right) and negative pairs (bottom-right). We then evaluate the integral of the product between the negative distribution and the cumulative density function for the positive distribution (shown with a dashed line), which corresponds to a probability that a randomly sampled positive pair has smaller similarity than a randomly sampled negative pair. Such histogram loss can be minimized by backpropagation. The only associated parameter of such loss is the number of histogram bins, to which the results have very low sensitivity.

(as histograms with linearly-interpolated values-to-bins assignments). In the second stage, the overlap between the two distributions is computed by estimating the probability that the two points sampled from the two distribution are in a wrong order, i.e. that a random negative pair has a higher similarity than a random positive pair. The two stages are implemented in a piecewise-differentiable manner, thus allowing to minimize the loss (i.e. the overlap between distributions) using standard backpropagation. The number of bins in the histograms is the only tunable parameter associated with our loss, and it can be set according to the batch size independently of the data itself. In the experiments, we fix this parameter (and the batch size) and demonstrate the versatility of the loss by applying it to four different image datasets of varying complexity and nature. Comparing the new loss to state-of-the-art reveals its favourable performance. Overall, we hope that the proposed loss will be used as an "out-of-the-box" solution for learning deep embeddings that requires little tuning and leads to close to the state-of-the-art results.

## 2 Related work

Recent works on learning embeddings use deep architectures (typically ConvNets [8, 10]) and stochastic optimization. Below we review the loss functions that have been used in recent works.

**Classification losses.** It has been observed in [8] and confirmed later in multiple works (e.g. [15]) that deep networks trained for classification can be used for deep embedding. In particular, it is sufficient to consider an intermediate representation arising in one of the last layers of the deep network. The normalization is added post-hoc. Many of the works mentioned below pre-train their embeddings as a part of the classification networks.

**Pairwise losses.** Methods that use pairwise losses sample pairs of training points and score them independently. The pioneering work on deep embeddings [3] penalizes the deviation from the unit cosine similarity for positive pairs and the deviation from $-1$ or $-0.9$ for negative pairs. Perhaps, the most popular of pairwise losses is the *contrastive* loss [5, 20], which minimizes the distances in the positive pairs and tries to maximize the distances in the negative pairs as long as these distances are smaller than some margin $M$. Several works pointed to the fact that attempting to collapse all positive pairs may lead to excessive overfitting and therefore suggested losses that mitigate this effect, e.g. a double-margin contrastive loss [12], which drops to zero for positive pairs as long as their distances fall beyond the second (smaller) margin. Finally, several works use non-hinge based pairwise losses such as log-sum-exp and cross-entropy on the similarity values that softly encourage the similarity to be high for positive values and low for negative values (e.g. [25, 28]). The main problem with pairwise losses is that the margin parameters might be hard to tune, especially since the distributions of distances or similarities can be changing dramatically as the learning progresses. While most works "skip" the burn-in period by initializing the embedding to a network pre-trained

for classification [25], [22] further demonstrated the benefit of admixing the classification loss during the fine-tuning stage (which brings in another parameter).

**Triplet losses.** While pairwise losses care about the absolute values of distances of positive and negative pairs, the quality of embeddings ultimately depends on the relative ordering between positive and negative distances (or similarities). Indeed, the embedding meets the needs of most practical applications as long as the similarities of positive pairs are greater than similarities of negative pairs [19, 27]. The most popular class of losses for metric learning therefore consider triplets of points $x_0, x_+, x_-$, where $x_0, x_+$ form a positive pair and $x_0, x_-$ form a negative pair and measure the difference in their distances or similarities. Triplet-based loss can then e.g. be aggregated over all triplets using a hinge function of these differences. Triplet-based losses are popular for large-scale embedding learning [4] and in particular for deep embeddings [13, 14, 17, 21, 29]. Setting the margin in the triplet hinge-loss still represents the challenge, as well as sampling "correct" triplets, since the majority of them quickly become associated with zero loss. On the other hand, focusing sampling on the hardest triplets can prevent efficient learning [17]. Triplet-based losses generally make learning less constrained than pairwise losses. This is because for a low-loss embedding, the characteristic distance separating positive and negative pairs can vary across the embedding space (depending on the location of $x_0$), which is not possible for pairwise losses. In some situations, such added flexibility can increase overfitting.

**Quadruplet losses.** Quadruplet-based losses are similar to triplet-based losses as they are computed by looking at the differences in distances/similarities of positive pairs and negative pairs. In the case of quadruplet-based losses, the compared positive and negative pairs do not share a common point (as they do for triplet-based losses). Quadruplet-based losses do not allow the flexibility of triplet-based losses discussed above (as they includes comparisons of positive and negative pairs located in different parts of the embedding space). At the same time, they are not as rigid as pairwise losses, as they only penalize the relative ordering for negative pairs and positive pairs. Nevertheless, despite these appealing properties, quadruplet-based losses remain rarely-used and confined to "shallow" embeddings [9, 31]. We are unaware of deep embedding approaches using quadruplet losses. A potential problem with quadruplet-based losses in the large-scale setting is that the number of all quadruplets is even larger than the number of triplets. Among all groups of losses, our approach is most related to quadruplet-based ones, and can be seen as a way to organize learning of deep embeddings with a quarduplet-based loss in an efficient and (almost) parameter-free manner.

## 3  Histogram loss

We now describe our loss function and then relate it to the quadruplet-based loss. Our loss (Figure 1) is defined for a batch of examples $X = \{x_1, x_2, \ldots x_N\}$ and a deep feedforward network $f(\cdot; \theta)$, where $\theta$ represents learnable parameters of the network. We assume that the last layer of the network performs length-normalization, so that the embedded vectors $\{y_i = f(x_i; \theta)\}$ are $L2$-normalized.

We further assume that we know which elements should match to each other and which ones are not. Let $m_{ij}$ be $+1$ if $x_i$ and $x_j$ form a positive pair (correspond to a match) and $m_{ij}$ be $-1$ if $x_i$ and $x_j$ are known to form a negative pair (these labels can be derived from class labels or be specified otherwise). Given $\{m_{ij}\}$ and $\{y_i\}$ we can estimate the two probability distributions $p^+$ and $p^-$ corresponding to the similarities in positive and negative pairs respectively. In particular $\mathcal{S}^+ = \{s_{ij} = \langle x_i, x_j \rangle \,|\, m_{ij} = +1\}$ and $\mathcal{S}^- = \{s_{ij} = \langle x_i, x_j \rangle \,|\, m_{ij} = -1\}$ can be regarded as sample sets from these two distributions. Although samples in these sets are not independent, we keep all of them to ensure a large sample size.

Given sample sets $\mathcal{S}^+$ and $\mathcal{S}^-$, we can use any statistical approach to estimate $p^+$ and $p^-$. The fact that these distributions are one-dimensional and bounded to $[-1; +1]$ simplifies the task. Perhaps, the most obvious choice in this case is fitting simple histograms with uniformly spaced bins, and we use this approach in our experiments. We therefore consider $R$-dimensional histograms $H^+$ and $H^-$, with the nodes $t_1 = -1, t_2, \ldots, t_R = +1$ uniformly filling $[-1; +1]$ with the step $\Delta = \frac{2}{R-1}$. We estimate the value $h_r^+$ of the histogram $H^+$ at each node as:

$$h_r^+ = \frac{1}{|\mathcal{S}^+|} \sum_{(i,j)\,:\,m_{ij}=+1} \delta_{i,j,r} \tag{1}$$

where $(i, j)$ spans all positive pairs of points in the batch. The weights $\delta_{i,j,r}$ are chosen so that each pair sample is assigned to the two adjacent nodes:

$$\delta_{i,j,r} = \begin{cases} (s_{ij} - t_{r-1})/\Delta, & \text{if } s_{ij} \in [t_{r-1}; t_r], \\ (t_{r+1} - s_{ij})/\Delta, & \text{if } s_{ij} \in [t_r; t_{r+1}], \\ 0, & \text{otherwise}. \end{cases} \qquad (2)$$

We thus use linear interpolation for each entry in the pair set, when assigning it to the two nodes. The estimation of $H^-$ proceeds analogously. Note, that the described approach is equivalent to using "triangular" kernel for density estimation; other kernel functions can be used as well [2].

Once we have the estimates for the distributions $p^+$ and $p^-$, we use them to estimate the probability of the similarity in a random negative pair to be more than the similarity in a random positive pair ( *the probability of reverse*). Generally, this probability can be estimated as:

$$p_{\text{reverse}} = \int_{-1}^{1} p^-(x) \left[ \int_{-1}^{x} p^+(y)\,dy \right] dx = \int_{-1}^{1} p^-(x)\,\Phi^+(x)\,dx = \mathbb{E}_{x \sim p^-}\left[\Phi^+(x)\right], \qquad (3)$$

where $\Phi^+(x)$ is the CDF (cumulative density function) of $p^+(x)$. The integral (3) can then be approximated and computed as:

$$L(X, \theta) = \sum_{r=1}^{R} \left( h_r^- \sum_{q=1}^{r} h_q^+ \right) = \sum_{r=1}^{R} h_r^- \phi_r^+, \qquad (4)$$

where $L$ is our loss function (the *histogram loss*) computed for the batch $X$ and the embedding parameters $\theta$, which approximates the reverse probability; $\phi_r^+ = \sum_{q=1}^{r} h_q^+$ is the cumulative sum of the histogram $H^+$.

Importantly, the loss (4) is differentiable w.r.t. the pairwise similarities $s \in \mathcal{S}^+$ and $s \in \mathcal{S}^-$. Indeed, it is straightforward to obtain $\frac{\partial L}{\partial h_r^-} = \sum_{q=1}^{r} h_q^+$ and $\frac{\partial L}{\partial h_r^+} = \sum_{q=r}^{R} h_q^-$ from (4). Furthermore, from (1) and (2) it follows that:

$$\frac{\partial h_r^+}{\partial s_{ij}} = \begin{cases} \frac{+1}{\Delta |\mathcal{S}^+|}, & \text{if } s_{ij} \in [t_{r-1}; t_r], \\ \frac{-1}{\Delta |\mathcal{S}^+|}, & \text{if } s_{ij} \in [t_r; t_{r+1}], \\ 0, & \text{otherwise}, \end{cases} \qquad (5)$$

for any $s_{ij}$ such that $m_{ij} = +1$ (and analogously for $\frac{\partial h_r^-}{\partial s_{ij}}$). Finally, $\frac{\partial s_{ij}}{\partial x_i} = x_j$ and $\frac{\partial s_{ij}}{\partial x_j} = x_i$. One can thus backpropagate the loss to the scalar product similarities, then further to the individual embedded points, and then further into the deep embedding network.

**Relation to quadruplet loss.** Our loss first estimates the probability distributions of similarities for positive and negative pairs in a semi-parametric ways (using histograms), and then computes the probability of reverse using these distributions via equation (4). An alternative and purely non-parametric way would be to consider all possible pairs of positive and negative pairs contained in the batch and to estimate this probability from such set of pairs of pairs. This would correspond to evaluating a quadruplet-based loss similarly to [9, 31]. The number of pairs of pairs in a batch, however tends to be quartic (fourth degree polynomial) of the batch size, rendering exhaustive sampling impractical. This is in contrast to our loss, for which the separation into two stages brings down the complexity to quadratic in batch size. Another efficient loss based on quadruplets is introduced in [24]. The training is done pairwise, but the threshold separating positive and negative pairs is also learned.

We note that quadruplet-based losses as in [9, 31] often encourage the positive pairs to be more similar than negative pairs by some non-zero margin. It is also easy to incorporate such non-zero margin into our method by defining the loss to be:

$$L_\mu(X, \theta) = \sum_{r=1}^{R} \left( h_r^- \sum_{q=1}^{r+\mu} h_q^+ \right), \qquad (6)$$

where the new loss effectively enforces the margin $\mu\,\Delta$. We however do not use such modification in our experiments (preliminary experiments do not show any benefit of introducing the margin).

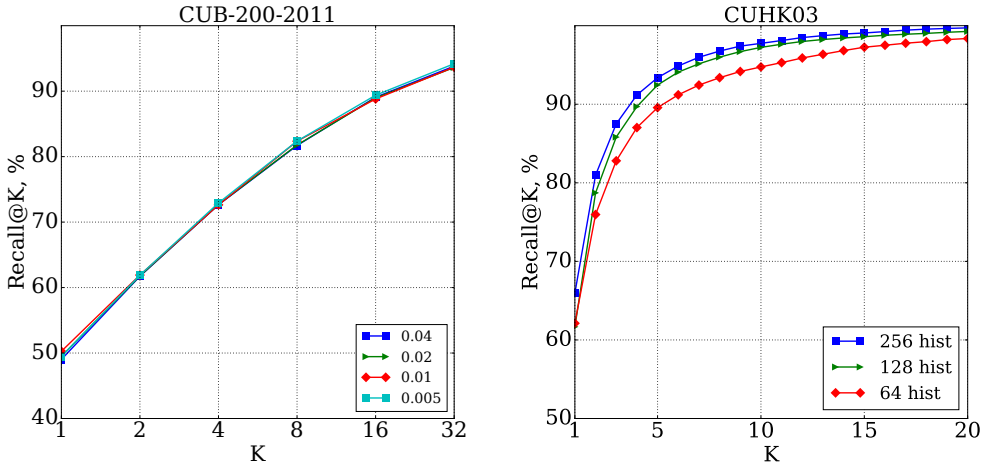

**Figure 2:** (left) - Recall@K for the CUB-200-2011 dataset for the Histogram loss (4). Different curves correspond to variable histogram step $\Delta$, which is the only parameter inherent to our loss. The curves are very similar for CUB-200-2011. (right) - Recall@K for the CUHK03 labeled dataset for different batch sizes. Results for batch size 256 is uniformly better than those for smaller values.

## 4   Experiments

In this section we present the results of embedding learning. We compare our loss to state-of-the-art pairwise and triplet losses, which have been reported in recent works to give state-of-the-art performance on these datasets.

**Baselines.** In particular, we have evaluated the Binomial Deviance loss [28]. While we are aware only of its use in person re-identification approaches, in our experiments it performed very well for product image search and bird recognition significantly outperforming the baseline pairwise (contrastive) loss reported in [21], once its parameters are tuned. The binomial deviance loss is defined as:

$$J_{dev} = \sum_{i,j \in I} w_{i,j} \ln(\exp^{-\alpha(s_{i,j}-\beta)m_{i,j}} + 1), \tag{7}$$

where $I$ is the set of training image indices, and $s_{i,j}$ is the similarity measure between $i$th and $j$th images (i.e. $s_{i,j} = cosine(x_i, x_j)$).

Furthermore, $m_{i,j}$ and $w_{i,j}$ are the learning supervision and scaling factors respectively:

$$m_{i,j} = \begin{cases} 1, \text{if } (i,j) \text{ is a positive pair,} \\ -C, \text{if } (i,j) \text{ is a negative pair,} \end{cases} \qquad w_{i,j} = \begin{cases} \frac{1}{n_1}, \text{if } (i,j) \text{ is a positive pair,} \\ \frac{1}{n_2}, \text{if } (i,j) \text{ is a negative pair,} \end{cases} \tag{8}$$

where $n_1$ and $n_2$ are the number of positive and negative pairs in the training set (or mini-batch) correspondingly, $\alpha$ and $\beta$ are hyper-parameters. Parameter $C$ is the negative cost for balancing weights for positive and negative pairs that was introduced in [28]. Our experimental results suggest that the quality of the embedding is sensitive to this parameter. Therefore, in the experiments we report results for the two versions of the loss: with $C = 10$ that is close to optimal for re-identification datasets, and with $C = 25$ that is close to optimal for the product and bird datasets.

We have also computed the results for the Lifted Structured Similarity Softmax (LSSS) loss [21] on CUB-200-2011 [26] and Online Products [21] datasets and additionally applied it to re-identification datasets. Lifted Structured Similarity Softmax loss is triplet-based and uses sophisticated triplet sampling strategy that was shown in [21] to outperform standard triplet-based loss.

Additionally, we performed experiments for the triplet loss [18] that uses "semi-hard negative" triplet sampling. Such sampling considers only triplets violating the margin, but still having the positive distance smaller than the negative distance.

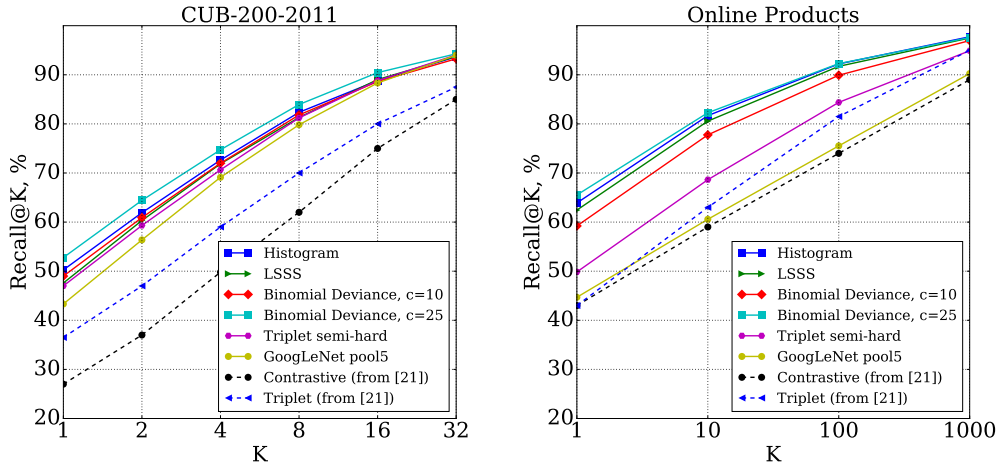

**Figure 3:** Recall@K for (left) - CUB-200-2011 and (right) - Online Products datasets for different methods. Results for the Histogram loss (4), Binomial Deviance (7), LSSS [21] and Triplet [18] losses are present. Binomial Deviance loss for $C = 25$ outperforms all other methods. The best-performing method is Histogram loss. We also include results for contrastive and triplet losses from [21].

**Datasets and evaluation metrics.** We have evaluated the above mentioned loss functions on the four datasets : CUB200-2011 [26], CUHK03 [11], Market-1501 [30] and Online Products [21]. All these datasets have been used for evaluating methods of solving embedding learning tasks.

The CUB-200-2011 dataset includes 11,788 images of 200 classes corresponding to different birds species. As in [21] we use the first 100 classes for training (5,864 images) and the remaining classes for testing (5,924 images). The Online Products dataset includes 120,053 images of 22,634 classes. Classes correspond to a number of online products from eBay.com. There are approximately 5.3 images for each product. We used the standard split from [21]: 11,318 classes (59,551 images) are used for training and 11,316 classes (60,502 images) are used for testing. The images from the CUB-200-2011 and the Online Products datasets are resized to 256 by 256, keeping the original aspect ratio (padding is done when needed).

The CUHK03 dataset is commonly used for the person re-identification task. It includes 13,164 images of 1,360 pedestrians captured from 3 pairs of cameras. Each identity is observed by two cameras and has 4.8 images in each camera on average. Following most of the previous works we use the "CUHK03-labeled" version of the dataset with manually-annotated bounding boxes. According to the CUHK03 evaluation protocol, 1,360 identities are split into 1,160 identities for training, 100 for validation and 100 for testing. We use the first split from the CUHK03 standard split set which is provided with the dataset. The Market-1501 dataset includes 32,643 images of 1,501 pedestrians, each pedestrian is captured by several cameras (from two to six). The dataset is divided randomly into the test set of 750 identities and the train set of 751 identities.

Following [21, 28, 30], we report Recall@K[1] metric for all the datasets. For CUB-200-2011 and Online products, every test image is used as the query in turn and remaining images are used as the gallery correspondingly. In contrast, for CUHK03 *single-shot* results are reported. This means that one image for each identity from the test set is chosen randomly in each of its two camera views. Recall@K values for 100 random query-gallery sets are averaged to compute the final result for a given split. For the Market-1501 dataset, we use the *multi-shot* protocol (as is done in most other works), as there are many images of the same person in the gallery set.

**Architectures used.** For training on the CUB-200-2011 and the Online Products datasets we used the same architecture as in [21], which conincides with the GoogLeNet architecture [23] up to the 'pool5' and the inner product layers, while the last layer is used to compute the embedding vectors. The GoogLeNet part is pretrained on ImageNet ILSVRC [16] and the last layer is trained from scratch. As in [21], all GoogLeNet layers are fine-tuned with the learning rate that is ten times less than

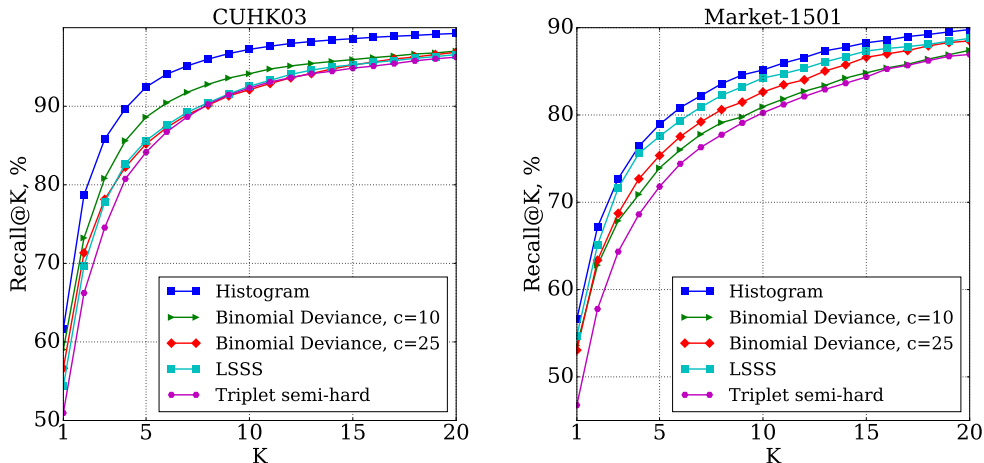

**Figure 4:** Recall@K for (left) - CUHK03 and (right) - Market-1501 datasets. The Histogram loss (4) outperforms Binomial Deviance, LSSS and Triplet losses.

the learning rate of the last layer. We set the embedding size to 512 for all the experiments with this architecture. We reproduced the results for the LSSS loss [21] for these two datasets. For the architectures that use the Binomial Deviance loss, Histogram loss and Triplet loss the iteration number and the parameters value (for the former) are chosen using the validation set.

For training on CUHK03 and Market-1501 we used the Deep Metric Learning (DML) architecture introduced in [28]. It has three CNN streams for the three parts of the pedestrian image (head and upper torso, torso, lower torso and legs). Each of the streams consists of 2 convolution layers followed by the ReLU non-linearity and max-pooling. The first convolution layers for the three streams have shared weights. Descriptors are produced by the last 500-dimensional inner product layer that has the concatenated outputs of the three streams as an input.

**Table 1:** Final results for CUHK03-labeled and Market-1501. For CUHK03-labeled results for 5 random splits were averaged. Batch of size 256 was used for both experiments.

| Dataset | r = 1 | r = 5 | r = 10 | r = 15 | r = 20 |
|---------|-------|-------|--------|--------|--------|
| CUHK03 | 65.77 | 92.85 | 97.62 | 98.94 | 99.43 |
| Market-1501 | 59.47 | 80.73 | 86.94 | 89.28 | 91.09 |

**Implementation details.** For all the experiments with loss functions (4) and (7) we used quadratic number of pairs in each batch (all the pairs that can be sampled from batch). For triplet loss "semi-hard" triplets chosen from all the possible triplets in the batch are used. For comparison with other methods the batch size was set to 128. We sample batches randomly in such a way that there are several images for each sampled class in the batch. We iterate over all the classes and all the images corresponding to the classes, sampling images in turn. The sequences of the classes and of the corresponding images are shuffled for every new epoch. CUB-200-2011 and Market-1501 include more than ten images per class on average, so we limit the number of images of the same class in the batch to ten for the experiments on these datasets. We used ADAM [7] for stochastic optimization in all of the experiments. For all losses the learning rate is set to $1e-4$ for all the experiments except ones on the CUB-200-2011 datasets, for which we have found the learning rate of $1e-5$ more effective. For the re-identification datasets the learning rate was decreased by 10 after the 100K iterations, for the other experiments learning rate was fixed. The iterations number for each method was chosen using the validation set.

**Results.** The Recall@K values for the experiments on CUB-200-2011, Online Products, CUHK03 and Market-1501 are shown in Figure 3 and Figure 4. The Binomial Deviance loss (7) gives the best results for CUB-200-2011 and Online Products with the $C$ parameter set to 25. We previously checked several values of $C$ on the CUB-200-2011 dataset and found the value $C = 25$ to be the optimal one. We also observed that with smaller values of $C$ the results are significantly worse than

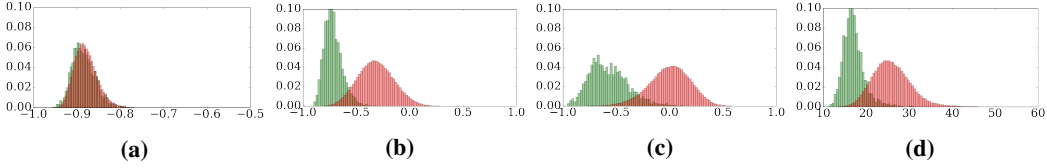

**Figure 5:** Histograms for positive and negative distance distributions on the CUHK03 test set for: (a) Initial state: randomly initialized net, (b) Network training with the Histogram loss, (c) same for the Binomial Deviance loss, (d) same for the LSSS loss. Red is for negative pairs, green is for positive pairs. Negative cosine distance measure is used for Histogram and Binomial Deviance losses, Euclidean distance is used for the LSSS loss. Initially the two distributions are highly overlapped. For the Histogram loss the distribution overlap is less than for the LSSS.

those presented in the Figure 3-left (for $C$ equal to 2 the best Recall@1 is 43.50%). For CUHK03 the situation is reverse: the Histogram loss gives the boost of 2.64% over the Binomial Deviance loss with $C = 10$ (which we found to be optimal for this dataset). The results are shown in the figure Figure 4-left. Embedding distributions of the positive and negative pairs from CUHK03 test set for different methods are shown in Figure 5b,Figure 5c,Figure 5d. For the Market-1501 dataset our method also outperforms the Binomial Deviance loss for both values of $C$. In contrast to the experiments with CUHK03, the Binomial Deviance loss appeared to perform better with $C$ set to 25 than to 10 for Market-1501. We have also investigated how the size of the histogram bin affects the model performance for the Histogram loss. As shown in the Figure 2-left, the results for CUB-200-2011 remain stable for the sizes equal to 0.005, 0.01, 0.02 and 0.04 (these values correspond to 400, 200, 100 and 50 bins in the histograms). In our method, distributions of similarities of training data are estimated by distributions of similarities within mini-batches. Therefore we also show results for the Histogram loss for various batch size values (Figure 2-right). The larger batches are more preferable: for CUHK03, Recall@K for batch size equal to 256 is uniformly better than Recall@K for 128 and 64. We also observed similar behaviour for Market-1501. Additionally, we present our final results (batch size set to 256) for CUHK03 and Market-1501 in Table 1. For CUHK03, Rekall@K values for 5 random splits were averaged. To the best of our knowledge, these results corresponded to state-of-the-art on CUHK03 and Market-1501 at the moment of submission. To summarize the results of the comparison: the new (Histogram) loss gives the best results on the two person re-identification problems. For CUB-200-2011 and Online Products it came very close to the best loss (Binomial Deviance with $C = 25$). Interestingly, the histogram loss uniformly outperformed the triplet-based LSSS loss [21] in our experiments including two datasets from [21]. Importantly, the new loss does not require to tune parameters associated with it (though we have found learning with our loss to be sensitive to the learning rate).

## 5   Conclusion

In this work we have suggested a new loss function for learning deep embeddings, called the Histogram loss. Like most previous losses, it is based on the idea of making the distributions of the similarities of the positive and negative pairs less overlapping. Unlike other losses used for deep embeddings, the new loss comes with virtually no parameters that need to be tuned. It also incorporates information across a large number of quadruplets formed from training samples in the mini-batch and implicitly takes into account all of such quadruplets. We have demonstrated the competitive results of the new loss on a number of datasets. In particular, the Histogram loss outperformed other losses for the person re-identification problem on CUHK03 and Market-1501 datasets. The code for Caffe [6] is available at: `https://github.com/madkn/HistogramLoss`.

**Acknowledgement:** This research is supported by the Russian Ministry of Science and Education grant RFMEFI57914X0071.

## Footnotes

[1]Recall@K is the probability of getting the right match among first K gallery candidates sorted by similarity.

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
