[Reviews · NeurIPS 2016]

Reviewer 1

Summary

The authors provide a new loss function for learning embeddings in deep networks, called histogram loss. This loss is based on a pairwise classification: whether two labels belong to the same class or not. In particular, the authors suggest to look at the similarity distribution of the embeddings on the L2 unit sphere (all embeddings are L2 normalized). The idea is to look at the distribution of the similar embedding (positive pairs) and the distribution of the non-similar ones (negative pairs) and make the probability that positive pairs has smaller score then negative pairs, smaller. After reviewing previous work in the area (Section 2), in Section 3 they develop a method how to estimate the Histogram loss. They begin with the definitions of Histogram (eq. 1, and 2) and based on them give the definition of $p_reverse$ which later we they want to minimize (eq.3). Afterward, they present an approximation to $p_reverse$ and show that it is differentiable. They conclude Section 3 with showing a connection to the quadruplet loss. In Section 4 the authors compare their method to other state of the art embedding loss, specifically, they consider the binomial deviance loss (eq. 7 and 8) and Lifted Structured Similarity Softmax (LSSS; eq. 9). They try these loses on 4 datasets: CUB200-2011 [25], CUHK03 [11], Market-1501 [29] and Online Products. In addition, different architectures for the neural network are used: For training on the CUB-200-2011 and the Online Products they use GoogleNet while for CUHK03 and Market-1501 they used the Deep Metric Learning architecture. The optimizatin algorithm used in this work is Adam (for stochastic optimization). They show the results using the Recall@K performance on all the datasets with different methods in Figures 3 and 4. The authors show how their method is sometime competitive and sometimes outperforms previous state of the art algorithms in the embeddings area.

Qualitative Assessment

I like and see the benefit of the authors' method for embeddings. The (almost) parameter free method is very appealing and have its benefits. The authors simply show the results of their algorithm and demonstrate that it works. Also, after many works where I had to read the supplementary material in order to understand the work, I appreciate the authors non-supplementary paper! Also, their throughout comparison to other methods in terms of loss functions, datasets, and different parameters, is very good. Also, I like the choice of the datasets, where there are many classes, which seems natural to this kind of loss. Two things bother me in this work: 1) The authors do not explain what is recall@K, where I needed to go back through the papers "Deep Metric Learning via Lifted Structured Feature Embedding" and then "Product quantization for nearest neighbor search". I could postulate what is the definition in this case is, but still it affected the self-contained nature of this work. 2) After doing so much profound work, it was not clear to me why each dataset has different set of graphs. I miss a good reason why such comparison is adequate in this context, especially in the light of that the authors did most of the work for throughout comparison. Typos: * In figure 5 caption, should it read |Red is for *positive* pair, green is for *negative* pairs?

Confidence in this Review

2-Confident (read it all; understood it all reasonably well)


Reviewer 2

Summary

This paper deals with an interesting problem with wide applicability, namely the design of cost functions for learning deep embeddings. After an overview of existing cost functions, the authors introduce a cost function which is based on the histograms of similarities between positive and negative pairs and which satisfies the important property that it is differentiable and allows for learning using back propagation.

Qualitative Assessment

In my opinion, the paper is well written and it presents an intuitive cost function that may end up being used by a large number of people. A minor weakness is that I think the authors could have done a better motivate the cost function from an intuitive perspective; it was easy to understand the underlying motivations for the design, but that was mainly because I think the cost functions makes a lot of sense. As for the experiments, I would say that they support the authors' statement that the proposed cost function may become the standard choice at some point in the future, but they also illustrate that the benefits are not substantial for all datasets. The paper also contains a small number of grammatical errors, but overall it was a pleasure to read the paper.

Confidence in this Review

2-Confident (read it all; understood it all reasonably well)


Reviewer 3

Summary

This paper present a new loss for learning deep embeddings, which is measured by the overlap between the distributions of similarities for positive and negative point pairs. Since this loss is differentiable, it is able to be backpropagated into a deep embedding network. The experimental results on several tasks, such as person re-identification, image search and fine-grained bird recognition, reveal the effectiveness of the new loss for embedding learning.

Qualitative Assessment

The proposed histogram loss is new and well-designed. It has several advantages, including: 1. It is piecewise-differentiable, so that it can be minimized by standard backpropagation. 2. It has only one tunable parameter which even is not sensitive. 3. Its computational complexity is relatively low, which makes learning more efficient. My concern about the paper includes three aspects: 1. The authors said they did not use equation (6) in their experiments. Why? Seems it is an interesting idea worthy of trying. 2. In the experiments, the batch size was fixed. As the distributions of similarities for positive and negative point pairs are computed in a batch, how do the results change by varying the batch size. 3. Since the main competitor, LSSS [21], also reports the results on CARS196, it’s better to test the proposed loss on it as well.

Confidence in this Review

2-Confident (read it all; understood it all reasonably well)


Reviewer 4

Summary

The paper has proposed a new loss function for learning deep embeddings, called the Histogram Loss. The new loss function is based on the idea of making the distributions of the similarities of the positive and negative pairs less overlapping. The news loss has virtually no parameters compared to other losses. The paper also demonstrated the competitive results of the new loss on a few datasets.

Qualitative Assessment

strengths of the paper: 1. The paper is well and clearly written. The authors have explained their methods clearly and have analysed the relation to quadruplet loss. The experiments setting are also explained in detail. 2. The paper has proposed a novel loss function call the Histogram Loss. The new loss function has virtually no parameters. An alternative way can achieve a similar goal using the quadruplet loss with an impractical complexity. The new loss brings down the complexity to quadratic in batch size. 3. The experiments have shown the competitive results of the new loss function. The paper has conducted intense experiments on a few datasets and is convincing enough to show consistent results of the new loss.

Confidence in this Review

2-Confident (read it all; understood it all reasonably well)


Reviewer 5

Summary

The paper addresses the problem of learning feature embeddings using deep learning approaches. Existing approaches can be grouped in base at which loss configuration of positive and negatives tuples they use to carry out the learning, ranging from pairs to quadruplets. The authors of this paper, instead proposed to consider a two stages loss: first, the similarity distributions between positive and negative pairs in the batch is computed; then the overlap between the two distribution is computed. The goal is to reduce the overlap between the two distributions. Advantageously, the proposed loss has only one tunable parameter (the number of bins of the histogram approximating the positive and negative distributions) to which it is quite robust as shown by the authors in Figure 2. From the point of view of results, the authors tested the proposed loss with three datasets: on two of them they showed lower performance w.r.t. state-of-the-art, on the other two (related to person re-identification) they showed the best performance.

Qualitative Assessment

I think that designing a loss that takes into account the statistics of the batch positive/negative examples rather than only pairs (resp. triplets, quadruplets) of them is a good idea. It also constitutes a more general approach that potentially encloses the pairs/triplets/quadruplets losses. The fact that there are basically no parameters to tune in the loss is a further advantage w.r.t. the pairs/triplets/quadruplets losses (the only parameter to tune is the number of bins in the histogram that approximate the positive/negative distributions). What is slightly less convincing to me are the experiment proposed. The authors tested the proposed loss on 4 datasets comprising online products and person re-identifications. On the first two dataset the proposed loss performs slightly worse that Binomial Deviance (BD) and similar to Lifted Structured Similarity Softmax (LSSS). It is not clear to me whether the parameters for the BD were tuned or not: only the results for C=25 and C=10 are shown. Therefore there is the chance that the BD performance could increases. On the two person re-identification datasets, instead histogram loss is outperforming the other two approaches. Figure 5 compares the positive and negative distributions after training. What is strange to me is the fact that the Histrogram loss does not separate the two populations significantly more than the other approaches. What I would like to see here is a figure showing how the distribution looks like before and after training.

Confidence in this Review

2-Confident (read it all; understood it all reasonably well)


Reviewer 6

Summary

This paper introduces a new differentiable loss for training deep metric embeddings. The main advantage of this loss is the absence of hyperparameters. In essence, given a sample batch, the loss estimates the probability that a negative sample pair's inner product is closer to a positive sample pair's inner product. Experiments were conducted on visual datasets and were evaluated using the Recall@K metric.

Qualitative Assessment

The loss function is intuitive and well-presented. In theory, this loss encourages the inner product between positive and negative sample pairs to be distinct, therefore effectively separating the embedding space into different classes. Experimental result shows that the histogram loss improves over many existing loss functions, but is outperformed by the binomial deviance loss on CUB-200-2011 and Online Products datasets. Many baseline results are missing for the CUHK03 and Market-1501 datasets.

Confidence in this Review

3-Expert (read the paper in detail, know the area, quite certain of my opinion)